# Tailoring of Ultrasmall NiMnO_3_ Nanoparticles: Optimizing Synthesis Conditions and Solvent Effects

**DOI:** 10.3390/molecules29204846

**Published:** 2024-10-13

**Authors:** Svetlana Saikova, Diana Nemkova, Anton Krolikov, Aleksandr Pavlikov, Mikhail Volochaev, Aleksandr Samoilo, Timur Ivanenko, Artem Kuklin

**Affiliations:** 1School of Non-Ferrous Metals, Siberian Federal University, Svobodny, 79, 660041 Krasnoyarsk, Russia; dsaykova@sfu-kras.ru (D.N.); akrolikov-cm19@stud.sfu-kras.ru (A.K.); apavlikov@sfu-kras.ru (A.P.); asamoylo@sfu-kras.ru (A.S.); 2Institute of Chemistry and Chemical Technology, Federal Research Center “Krasnoyarsk Science Center of the Siberian Branch of the Russian Academy of Sciences”, Akademgorodok, 50/24, 660036 Krasnoyarsk, Russia; timivonk@gmail.com; 3Kirensky Institute of Physics, Federal Research Center “Krasnoyarsk Science Center of the Siberian Branch of the Russian Academy of Sciences”, Akademgorodok, 50/38, 660036 Krasnoyarsk, Russia; volochaev@iph.krasn.ru; 4Department of Physics and Astronomy, Uppsala University, P.O. Box 516, SE-751 20 Uppsala, Sweden; artem.kuklin@physics.uu.se

**Keywords:** nickel manganese oxide, nanoparticles, ultrasonic treatment, N-methyl-2-pyrrolidone, dimethyl sulfoxide, dimethylformamide

## Abstract

Nickel manganese oxide (NiMnO_3_) combines magnetic and dielectric properties, making it a promising material for sensor and supercapacitor applications, as well as for catalytic water splitting. The efficiency of its utilization is notably influenced by particle size. In this study, we investigate the influence of thermal treatment parameters on the phase composition of products from alkali co-precipitation of nickel and manganese (II) ions and identify optimal conditions for synthesizing phase-pure nickel manganese oxide. Ultrafine nanoparticles of NiMnO_3_ (with sizes as small as 2 nm) are obtained via liquid-phase ultrasonic dispersion, exhibiting a narrow size distribution. A systematic exploration of the solvent nature (water, N-methyl-2-pyrrolidone, dimethyl sulfoxide, dimethylformamide) on the efficiency of ultrasonic dispersion of NiMnO_3_ nanoparticles is provided. It is demonstrated that particle size is influenced not only by absorbed acoustic power, dependent on the physical properties of the used solvent (boiling temperature, gas solubility, viscosity, density) but also by the chemical stability of the solvent under prolonged ultrasonic treatment. Our findings provide insights for designing ultrasonic treatment protocols for nanoparticle dispersions with tailored particle sizes.

## 1. Introduction

Nickel manganese oxide (NiMnO_3_) possesses an ilmenite-type structure (spatial symmetry group $R\bar{3}$, rhombohedral unit cell), where oxygen atoms form a dense hexagonal packing. Two-thirds of the octahedral sites are occupied by metal atoms (Ni and Mn), while the remaining vacancies (one-third) are unoccupied. Nickel and manganese are distributed in the structure by layers: O–Mn–O–Ni–O–Mn, and so forth (see Figure 1). The material exhibits semiconducting properties (band gap of 2.71 eV [1]), shows decent mechanical strength characteristics (bulk modulus of elasticity is 217 GPa [2]), and possesses ferrimagnetism where the magnetic moments of the Mn^4+^ and Ni^2+^ sublattices are oriented in opposite directions to each other.

Nickel manganese oxide is classified as a multiferroic material, combining magnetic and dielectric properties. Multiferroics are renowned for their ability to undergo ionic displacements within the crystal lattice when subjected to external magnetic fields. This phenomenon makes NiMnO_3_ a versatile candidate for catalyzing water oxidation [3,4], ozone decomposition [5] to generate oxygen, and applications in supercapacitors [6,7]. Moreover, in the form of a nanocomposite with Mn_2_O_3_, nickel manganese oxide serves as an efficient anode material in lithium-ion batteries, showing high discharge/charge capacities and remarkable cycling stability [8].

Critical to its diverse applications are the morphology, particle size, and nanostructure of NiMnO_3_. Nanomaterials with enhanced specific surface areas often outperform their bulk counterparts, with monodisperse particles exhibiting particularly desirable characteristics [9]. Achieving these attributes requires a methodical selection of synthesis techniques and precise control over process parameters. Common approaches for NiMnO_3_ nanoparticle synthesis encompass hydrothermal methods [6,10,11,12], alkaline co-precipitation [13], sol–gel processes [14], and others.

Among these methods, liquid-phase ultrasonic dispersion stands out as a promising avenue for nanoparticle fabrication [15,16,17,18]. Typically employing ultrasound frequencies ranging from 20 to 100 kHz, this technique minimizes the adverse effects of vapor cavitation at lower frequencies while avoiding excessive cavitation thresholds at higher frequencies, thereby enhancing the efficiency of sonochemical processes.

When ultrasound traverses through a liquid medium, it induces mechanical oscillations within the fluid, generating acoustic streams therein. If the liquid medium contains dissolved gas, gas bubbles undergo several cycles of compression and expansion until they collapse [19]. During bubble oscillations and collapse, various physical effects occur, including shock waves, microjets, turbulence, shear forces, and more. Studies have demonstrated that spherical shock waves exert the most significant influence on solid materials dispersed in the medium [20,21,22]. Additionally, bubble collapse leads to a momentary rise in local temperature (reaching 5000 K) and pressure (up to 2000 atm) at extremely high rates (heating and cooling rates reaching 10^10^–10^12^ degrees Celsius per second), inducing extreme conditions that can result in plasma formation and chemical transformations of substances within the gas bubble [23,24,25,26]. For instance, in water as the liquid medium, ultrasound can trigger its homolysis, producing radicals such as H• and OH•.

The probability of cavitation bubble formation (cavitation threshold), their stability, and the acoustic power absorbed by the liquid phase are determined by the physical properties of the solvent: density, viscosity, boiling temperature, heat capacity, gas solubility, surface tension, etc., [27,28,29]. Liquids with low viscosity (indicative of weak intermolecular interactions) and low surface tension facilitate the creation of acoustic cavitation with high intensity, resulting in a low cavitation threshold. Furthermore, the presence of any free surfaces, cracks, or defects in solid particles within the liquid medium can significantly reduce the cavitation threshold, serving as cavitation-bubble nucleation sites [30]. Given that the nature of the liquid phase often dictates the effectiveness of sonochemical processes, selecting an appropriate solvent can control ultrasonic dispersion and guide it toward obtaining products with desired characteristics.

Despite the stochastic nature and complexity of ultrasound cavitation mechanisms, the regularities of ultrasound dispersion of solid materials are not yet fully understood. While some experimental works have explored the selection of effective solvents for solid dispersion [20,21], systematic studies analyzing the influence of dispersion medium parameters on the size, morphology, and properties of obtained nanoparticles remain scarce. This study aims to investigate the impact of solvent physical properties (distilled water, N-methyl-2-pyrrolidone (NMP), dimethyl sulfoxide (DMSO), and dimethylformamide (DMF)) on the absorbed acoustic power and the effectiveness of ultrasonic dispersion of nickel manganese oxide nanoparticles, as well as the size and morphology of the resulting nanomaterials.

## 2. Results and Discussion

### 2.1. Fabrication and Characterization of Nickel Manganese Oxide

Nickel Manganese Oxide was synthesized by coprecipitation of Ni(II) and Mn(II) salts (pH 10, NaOH) at 25 °C for 1 h, followed by several rinsings and drying at 80 °C.

The as-synthesized samples exhibited a crystalline nature and, according to the X-ray diffraction (XRD) data (Figure 2b), contained MnOOH—57.0 wt.%, Mn_3_O_4_—33.0 wt.%, and Ni(OH)_2_×NiOOH—10.0 wt.% (the non-stoichiometric nickel content is attributed to its partial presence in the amorphous phase, see Table 1).

Figure 2a displays the thermogram of the precipitation product. Several thermal effects are observed on the DSC curve, associated with the decomposition of the initial manganese and nickel hydroxides (at 130.1 and 268.7 °C, respectively). Of particular interest is the exothermic effect, with a maximum at 344.7 °C, which presumably corresponds to the formation process of NiMn_2_O_4_ [5]. At 771.4 °C, an endothermic effect is observed, accompanied by a 4.6% decrease in mass, likely corresponding to the decomposition of NiMnO_3_ with the release of oxygen. To refine the thermal analysis data, the obtained samples were subjected to heat treatment at various temperatures: 450 °C, and 600 °C, for 3–11 h, and the resulting products were examined by X-ray phase analysis (Figure 2, Table 1).

Analysis of the XRD data shows a complex evolution of the composition of the calcination products with temperature and time of heat treatment. The sample calcined for 3 h at 350 °C still represents a mixture of Mn_3_O_4_ and NiO oxides. These phases begin to interact with each other, forming the NiMn_2_O_4_ spinel above 400 °C (the discrepancy with the thermal analysis data is likely due to differences in the conditions of heat treatment). Calcination at 450 °C leads to the formation of spinel (62%). The sample also contains NiMnO_3_ (23%) and nickel oxide (15%). A further increase in the calcination temperature to 600 °C results in the decomposition of the previously formed NiMn_2_O_4_ to NiMnO_3_. At temperatures above 600 °C, NiMnO_3_ in turn transforms into a cubic spinel and a mixture of Mn_3_O_4_, Mn_2_O_3_, and MnO_2_ oxides [31,32,33].

With an increase in the calcination time (at 450 °C) to 8 h, there is a gradual increase in the yield of the NiMnO_3_ phase, to 100%. Subsequently (11 h), NiMnO_3_ decomposes to form NiMn_2_O_4_. At a higher temperature (600 °C), the processes of formation and decomposition of complex oxide phases occur faster, and the range of existence of a single-phase product was not observed by us. Based on the conducted research, optimal conditions for the thermal treatment of the precipitation product were determined: calcination temperature of 450 °C, and calcination time of 8 h. The results of the structure refinement of the sample synthesized under optimal conditions using the Rietveld method are given in Table 2 and Appendix A.

Figure 3a shows the TEM image of NiMnO_3_ NPs synthesized under optimal conditions and their size distribution histograms (Figure 3b–d). TEM analysis demonstrates the particles with different shapes: spheres and rods. The median size of the spherical nanoparticles is about 15 nm (Figure 3b), while the width of the nanorods is 10 nm (Figure 3c) and the length is 39 nm (Figure 3d). The average nanoparticle crystallite size was determined from the peak broadening for the four most intensive powder XRD peaks using the Scherrer equation, and it was 9.3 ± 0.9 nm. According to SEM data (3e), fine particles form irregular-shaped aggregates.

Figure 3f shows the UV-Vis spectra of NiMnO_3_ nanoparticles. It can be observed that the material absorbs across the entire visible spectrum, with a gradual increase in optical density from 320 nm to 1000 nm.

### 2.2. Ultrasonic Dispersion of Nickel Manganese Oxide

The ultrasmall nickel manganese oxide nanoparticles were obtained by sonication of NiMnO_3_ during 50 h in dimethylformamide, dimethylsulfoxide, N-N-Methyl-2-pyrrolidone and distilled water. All these solvents have different boiling temperatures (physical properties of the solvents are given in Appendix A). The obtained organosols or the hydrosol were centrifuged for 15 min at 15,000 r/min, to precipitate larger particles. The separated supernatants were examined using transmission electron microscopy (TEM) (Figure 4). It is evident that the particle size depends on the solvent used. For instance, in water, irregularly shaped large nanoparticles (84 ± 2 nm) were obtained. Conversely, organic solvents (dimethylformamide, dimethyl sulfoxide, and N-Methyl-2-pyrrolidone) yielded ultrasmall spherical nanoparticles (2–5 nm) with a narrow size distribution (particle-size distribution histograms are shown in Figure 4b,d,f,h). The smallest particles of 1.9 ± 0.1 nm (range 0.7–3.1 for 0. 95 probability) were obtained in N-Methyl-2-pyrrolidone, while particles with sizes of 3.7 ± 0.2 nm (range 1.6–5.8 for 0. 95 probability) and 4.2 ± 0.2 nm (range 1.8–6.6 for 0. 95 probability) were exfoliated in dimethyl sulfoxide and dimethylformamide, respectively.

The following methods are commonly employed to produce ultrafine particles: laser chemical vapor deposition (LCVD), ion-beam synthesis (IBS) and electrostatic adsorption with high-temperature treatment and hydrogen atmosphere processing [34,35,36]. These methods are energy-intensive, have high production costs, require advanced instrumentations and can be applied to obtain a narrow range of metals and their compounds. Ultrasonic dispersion is a common method for obtaining nanosheets of van der Waals (VdW) compounds, particularly metal sulfides. In [18], the authors employed ultrasonic treatment (20 kHz, 2000 W) of WS_2_ nanosheets, which were subsequently subjected to quantum dot production via boiling the nanosheets in ethylene glycol for 96 h. The size of QD varied from 1 to 3 nm. This technique has only been applied to VdW compounds, and furthermore, it is more labor- and energy-intensive than the one used in our study.

Cavitation can be characterized by three main parameters: the cavitation inception threshold, cavitation intensity, and stability of the cavitating bubbles and flows. The cavitation inception threshold refers to the minimum intensity of ultrasonic waves at which the cavitation process begins. This value depends on several parameters characterizing both the liquid state and the acoustic field. An increase in the cavitation threshold occurs with increased viscosity, surface tension, and gas content in the liquid, as well as with higher frequencies of acoustic oscillations and shorter pulse durations [30,37,38,39]. During the rarefaction phase of the acoustic wave, gases dissolved in the liquid form small bubbles, which can act as cavitation nuclei, thereby significantly a reduction of the cavitation inception threshold. Solid impurities in the liquid also have an effect. Cavitation intensity refers to the volume-averaged power of spherical shock waves generated in the liquid medium upon the “collapse” of cavitation bubbles. Stationary cavitation bubbles are characterized by a relatively long lifetime and high vapor pressure in the cavities. Such bubbles are formed at sufficiently low ultrasonic intensities. Transient cavitation (acoustic energy density exceeding 10 W/cm^2^) is characterized by high intensity, as the vapor pressure in the bubble cavities is very low, due to their short lifetime. One of the primary indicators of cavitation efficiency is the absorbed acoustic power (AAP) [30,40]. It is also dependent on the physical properties of the solvent, such as density, specific heat capacity, and viscosity, and can be determined using Equation (1), see Section 3.5.

The absorbed acoustic power (Table 3) increases in the series DMFA < DMSO < H_2_O < NMP. Despite the comparatively higher acoustic power absorbed by water, particle dispersion in that medium occurs to a lesser extent. This phenomenon can be attributed to the low boiling temperature of water in comparison to other solvents. This results in an increase in evaporation, which subsequently raises the total vapor pressure within the bubble. Consequently, a notable decrease in cavitation intensity is observed due to the transition from gaseous to vaporous cavitation, resulting in a significant reduction in cavitation collapse efficiency. Additionally, the high surface tension of water markedly elevates the cavitation onset threshold and diminishes the rate of sonochemical processes [21,30].

In the case of organic solvents, the absorbed acoustic power is inversely proportional to the size of the obtained ultrafine nanoparticles. A similar trend was observed during ultrasonic dispersion of TiO_2_, Al_2_O_3_, and SiO_2_ in water [40]. When using NMP, a change in the solution’s color was observed after merely 15 h, which indicated a higher level of ultrasonic dispersion efficiency within this solvent.

It was established by Suslick et al. [41] that all organic liquids generate free radicals upon ultrasonic irradiation. Furthermore, prolonged ultrasonic treatment leads to chemical changes and degradation of solvents, induced by both ultrasound and heating [25,42,43]. It is possible that solvent degradation products may undergo chemical reactions with dispersed particles, thereby affecting the ultrasonic treatment process. Nepal et al. [44] noted that the oxidation of NMP enhances the efficiency of sonochemical reactions, as the formed reactive N- and ω-hydroperoxides oxidize the surface layer of dispersed particles, leading to its exfoliation without aggressive mechanical treatment. Moreover, the oxidation products of NMP, along with its polymerization, contribute to the adsorption on the surface of the particles. This adsorption reduces the surface energy and increases surface charge, thereby aiding in stabilization of the nanoparticles.

Following ultrasonic dispersion, minor alterations are observed in the IR spectra of DMSO (around 1200 cm^−1^) and NMP (around 3400–3500 cm^−1^) (Figure 5), which we attribute to partial solvent degradation. Specifically, in the case of NMP, an increase in the intensity of the absorption band in the region 3400–3500 cm^−1^ is observed after sonication for 50 h, possibly due to the appearance of a N-H bond resulting from solvent ring opening [45]. Weak peaks with maxima at 1123 cm^−1^ (asymmetric O=S=O) [46] and 1210 cm^−1^ (stretching sulfonic acid group) [47] are observed in the case of DMSO, likely indicating the oxidation of DMSO into dimethyl sulfone.

The changes in N-methyl-2-pyrrolidone during ultrasonic treatment are further confirmed by NMR spectroscopy data. In the 1H NMR spectrum of NMP (Figure 6a), four signals are present, one corresponding to the methyl group on the nitrogen atom, and three others to methylene groups of the ring. The signals of CH_2_ groups are broadened due to line overlapping arising from the geometric non-equivalence of CH_2_ protons and their spin–spin interactions with each other. In the 13C NMR spectrum (Figure 6b), three signals of CH_2_ groups (17.75, 28.85, and 48.87 ppm), a signal of the methyl group at 30.38 ppm, and a signal of C=O located in the deshielded region of the spectrum at 173.74 ppm are observed.

The determination of NMP transformation products under ultrasound was conducted using correlation NMR experiments (2D-1H-1H-DQF-COSY, 2D-1H-13C-HMBC, and 2D-1H-13C-HSQC) (Figure 6c,d). In the low-field region, a well-resolved signal characteristic of the carboxyl group proton (11.9 ppm) is observed, which is also identified in the 13C spectrum (174.80 ppm). Signals of 1H and 13C with multiple bonds are located in characteristic regions of the spectra (5–7 ppm for 1H and 70–100 ppm for 13C). The signals of CH_2_ and CH_3_ groups overlap with signals of the original NMP; however, they can be identified through 2D correlation spectra, from which it follows that the main product of NMP degradation is 4-methylaminobut-3-enoic acid.

The analysis of the absorption spectra of supernatants containing nickel manganese oxide nanoparticles (denoted as nps), as well as solvents subjected to two-hour ultrasonic (denoted as US) and thermal treatment at 90 °C (denoted as H), (Figure 7) indicates a gradual accumulation of solvent degradation products with increasing duration of ultrasonic treatment. Following a mere two hours of treatment, a change in color from transparent to yellow was observed in pure solvents. This indicated the onset of chemical changes in them, which further intensify, leading to the absorption of electromagnetic radiation over a wide range of wavelengths: 300–450 nm for DMFA, DMSO, and water, and 300–750 nm for NMP. The nanoparticles also demonstrate light absorption in these ranges.

The altered substances are capable of chemically interacting with the dispersed material and, consequently, affecting the ultrasonic treatment process. To exclude the effect of changes occurring in the solvent during sonication and to evaluate the influence of the nature and properties of the solvent, it is necessary to reduce the duration of ultrasonic interaction. However, as previously established [48], the primary mechanism of crystal destruction is fatigue, which is initiated by numerous interactions between the solid phase and shock waves propagating from imploding bubbles. This process typically results in the growth of an existing crack to a critical size, leading to brittle fracture. In the absence of cracks, bond rupture may occur at any point within the crystal, depending on the number of internal defects, which act as stress amplifiers and crack-initiation sites. Since solid-phase particles continuously circulate in the solution during sonication, changing their orientation, they will be subjected to less effective action, and prolonged time will be required for their dispersion. It was demonstrated in [22] that cavitation bubbles with a radius of 50–80 μm cause damage to the adjacent crystal after a period of 1680 acoustic cycles. In order to reduce the ultrasonic treatment time, the use of suspended particles in the solvent was replaced by a fixed-volume sample, which did not participate in the medium movement. Thus, the influence of the solvent nature on the effectiveness of ultrasonic dispersion was studied on a model system, which consisted of a thin aluminum foil which was fixed near the bottom of the glass, as shown in Appendix A.

During the experiment, the sample was destroyed, with the destruction occurring directly under the ultrasonic probe (Appendix A). This was accompanied by a decrease in the mass of the sample and the formation of fumes (for clarity, photographs demonstrating the Tyndall effect are given in Figure 8c–f). The histogram in Figure 8b illustrates the mass loss of the foil during 15 min of ultrasonic treatment, as a function of the nature of the solvent. The lowest efficiency of ultrasonic treatment, as in the case of nickel manganese oxide dispersion, was observed for water, which, as noted above, is associated with a high cavitation threshold and low intensity, due to high water-vapor pressure. The obtained results do not support the conclusions made in [20,21] that water is an effective solvent for sonochemical treatment. However, they are consistent with the data [30] which demonstrate the effectiveness of solvents with high boiling points. The efficiency of ultrasonic dispersion in DMFA and DMSO does not depend on the nature of the solvent within the experimental error. At the same time, in the electronic spectrum of dimethyl sulfoxide after sonication, significant absorption in the range from 300 to 600 nm is observed, which can be explained by greater stabilization of aluminum particles in DMSO. The test system showed a slightly lower efficiency of ultrasonic dispersion in NMP, which can be explained by the maximum boiling temperature of NMP (202 °C) for all the studied solvents, which slightly increases the cavitation onset threshold due to its low vapor pressure. The observed outcome differs from that obtained for NiMnO_3_, thereby providing evidence that the chemical transformations that occurred in NMP during its prolonged ultrasonic treatment contribute to the dispersion of solid materials.

## 3. Materials and Methods

### 3.1. Chemicals

Nickel nitrate (Ni(NO_3_)_2_·6H_2_O), manganese sulfate (MnSO_4_·5H_2_O), sodium hydroxide (NaOH), 35% hydrogen peroxide (H_2_O_2_), sodium dodecyl sulfate (CH_3_(CH_2_)_11_OSO_3_Na), N-Methyl-2-pyrrolidone (C_5_H_9_NO, Sisco Research Laboratories Pvt. Ltd., Mumbai, India, 99.5%), dimethyl sulfoxide ((CH_3_)_2_SO, Acros Organics BVBA, Brussel, Belgium, 99.7%), dimethylformamide (DMF, (CH_3_)_2_NC(O)H), Al foil (width 9.5 µm, GOST 745-2014, RUSAL-Sayanal, Sayanogorsk, Russia), and other chemicals were of analytical grade, were purchased from Sigma-Aldrich, and were used as received.

### 3.2. Synthesis of Nickel Manganese Oxide Nanoparticles

In this study, NiMnO_3_ nanoparticles were synthesized by the alkaline coprecipitation method. In a typical procedure, 50 mL Ni(NO_3_)_2_ (0.4 mol/L) was mixed with 50 mL MnSO_4_ (0.4 mol/L), 96 mL sodium hydroxide (1 mol/L, excess 20%), and 1.5 mL hydrogen peroxide (35%). Subsequently, the precipitate was repeatedly washed with water until a neutral pH and a negative reaction for sulphate and nitrate ions were achieved. The mixture was stirred (1000 rpm) for 1 h at 25 °C using a shaking water bath. The precipitate was centrifuged, washed with distilled water, dried in air at 80 °C for 1 h, and finally annealed in a muffle furnace for 3–11 h at 350 °C, 450 °C or 600 °C.

### 3.3. Ultrasonic Dispersion of Nickel Manganese Oxide

NiMnO_3_ ultrasmall nanoparticles were obtained by sonication of the as-synthesized nickel manganese oxide. For this purpose, 30 mL of organic solvent (NMP, DMSO, DMF) or distilled water was added to 100 mg of nickel manganese oxide, and sonicated for 50 h using the ultrasonic device “Volna” with titanium probe (22 kHz, 400 W, Center of Ultrasonic Technologies LLC, Biysk, Russia). The solvent vessel was cooled on an ice bath with a volume of 2 L to prevent any temperature increase of the sample. The total number of samples per treatment was 3. Throughout the experiment, the temperature inside the vessel did not exceed 50 °C. To prevent solvent evaporation, the vessel was covered with a polyethylene film. Additionally, a constant volume of solvent was maintained by periodic addition to the reaction medium. The resulting solution was then centrifuged for 15 min at 14,000 rpm (13,400× *g*), and the supernatant obtained after separation was analyzed using physical methods.

### 3.4. Investigation of the Solvent Effect on the Aluminum Ultrasonic Treatment

In a typical experiment, a pre-weighed square sheet of aluminum foil (7 × 7 cm, with a width of 9.5 µm) was positioned near the bottom of a 50 mL glass beaker. Subsequently, 50 mL of solvent (H_2_O/DMF/DMSO/NMP) was introduced into the beaker, and an ultrasonic probe was lowered to within 1 cm from the bottom of the beaker (see Appendix A), then subjected to sonication for 15 min at a power of 400 W. Following the experiment, the foil sheet was extracted, rinsed, dried in an oven at 110 °C, and then re-weighed. The resulting sols, containing aluminum particles generated during ultrasonic treatment, underwent spectrophotometric and dynamic light scattering analyses.

### 3.5. Determining the Specific Absorbed Acoustic Power

The investigated solvent was placed in a thermoresistant glass beaker with a capacity of 50 mL, which was then placed into a calorimeter and thermostated at a temperature of 25 °C. The contents of the calorimeter were sonicated for 15 s (at a power of 400 W) using a “Volna” apparatus equipped with a titanium probe (22 kHz, 400 W, Center of Ultrasonic Technologies LLC, Biysk, Russia), and the change in temperature was recorded. The absorbed acoustic power (in W∙cm^−2^) in the volume of the liquid was calculated using the formula [40]:P = (Cp × ∆T × m)/(V × τ)(1)
where C_p_ denotes the specific heat capacity of the liquid, expressed in J/(kg∙K); ∆T represents the experimentally determined change in temperature in the liquid over a period of τ (s); m denotes the mass (g), while V refers to the volume (m^3^) of the liquid being processed.

### 3.6. Nanoparticle Characterization

Thermal analysis (TA) was conducted using a synchronous thermal analyzer SDT Q600, coupled with an IR Fourier spectrometer Nicolet 380s TGA/FT-IR interface (gas-phase analysis attachment). Thermograms were acquired during heating at a rate of 20 °C/min in an air atmosphere, with an airflow rate of 50 mL/min.

X-ray powder diffraction (XRD) data were collected using a Shimadzu XDR-600 diffractometer (Shimadzu Corporation, Kyoto, Japan) with CuKα radiation. The sample was ground in an agate mortar and loaded directly into the sample holder. Scanning parameters included a range from 5 to 70° on the 2θ scale with a step size of 0.013° and a time interval of 50 s/step, conducted in air at room temperature. Approximately 0.2 g of the substance was loaded into an alumina sample holder. Phase identification was performed using the PDF-2 database.

Transmission electron microscopy (TEM) analysis was conducted using a Hitachi 7700M microscope (Hitachi Corporation, Tokyo, Japan) at an accelerating voltage of 100 kV. The STATISTICA package was used for statistical data processing. The particle-size distribution histograms were obtained from more than 300 particles.

FTIR spectra were recorded using a Bruker Tensor 27 FTIR spectrometer (Bruker, Bremen, Germany) over the range of 4000–400 cm^−1^. UV–vis absorption spectra were obtained using a GENESYS 10S UV–Vis spectrophotometer (Thermo Scientific, Bedford MA, USA) with a glass cell having an optical path length of 1 cm, covering the range from 200 to 1100 nm.

Scanning electron microscopy (SEM) micrographs of the original nickel manganese oxide were obtained using a JSM-7001F scanning electron microscope (JEOL Ltd., Tokyo, Japan) at an accelerating voltage of 15 kV.

Infrared (IR) spectra were recorded using a multiple internal reflection (MIR) attachment of the Fourier-transform infrared (FTIR) spectrometer Bruker Tensor 27 (Bruker, Germany). Initially, the background spectrum was obtained from a crystal made of ZnSe. Then, the sample was placed on the crystal to cover its entire surface, and the sample spectrum was recorded. The spectral range for ZnSe crystal was 4000–400 cm^−1^, with a resolution of 4 cm^−1^ and 32 scans. The resulting spectrum was obtained by subtracting the background spectrum from the sample spectrum. Spectra were processed using the OPUS 7.5 software package (Bruker).

The hydrodynamic diameter of particles was determined by dynamic light scattering using a Zetasizer NanoZS instrument (Malvern Instruments LTD, Malvern, United Kindom) with a laser wavelength of 632.8 nm and a scattering angle of 173°. One milliliter of the sol was transferred to a glass cuvette (l = 1 cm).

Nuclear magnetic resonance (NMR) spectra were recorded on a Bruker AVANCE 400 spectrometer (Bruker, Germany) using standard 5 mm NMR tubes. ^1^H spectra were acquired using a single pulse at the operating frequency of 600 MHz with a relaxation delay of 5 μs. Water suppression was achieved using a standard zgpr pulse sequence from the Bruker library. ^13^C{^1^H} spectra with proton decoupling were recorded at the operating frequency of 150 MHz with a relaxation delay of 6.5 μs, accumulating 512 scans over 19 h. Chemical shifts were referenced to the signal of tetramethylsilane (TMS) set at 0 ppm. All spectra were processed using the Topspin 3.2 software package.

## 4. Conclusions

In this work we defined the optimal synthesis conditions for nickel manganese oxide nanoparticles, achieving sizes of 10 nm (for spherical particles) and 10 × 39 nm (for rods), which could be utilized in sensor and supercapacitor applications. Ultrasonic dispersion of the synthesized nickel manganese oxide was conducted in various solvents: dimethyl sulfoxide (DMSO), dimethylformamide (DMFA), N-methyl-2-pyrrolidone (NMP), and water (H_2_O). It was demonstrated that particle size depended on the solvent nature. In water, large irregularly shaped nanoparticles (84 ± 2 nm) were obtained, while organic solvents yielded ultrasmall particles with narrow size distributions: 1.9 ± 0.1 nm for NMP, 3.7 ± 0.2 nm for DMSO, and 4.2 ± 0.2 nm for DMFA.

To assess the influence of solvents on cavitation efficiency, the absorbed acoustic power was determined. It was shown that this parameter only considered the energy component of the process, neglecting the high cavitation threshold in systems with significant surface tension and the transition of cavitation mechanism from liquid to gas in cases of high vapor-pressure solvents with low boiling points. Fourier-transform infrared spectroscopy (FTIR), UV–Vis spectroscopy, and nuclear magnetic resonance spectroscopy (NMR) revealed that dimethyl sulfoxide and N-methyl-2-pyrrolidone underwent thermal and sonochemical decomposition and oxidation during prolonged treatment, affecting the efficiency of ultrasonic dispersion.

A model system (a fixed aluminum-foil sample) and experimental design were proposed to study the solvent influence on cavitation efficiency, enabling a significant reduction in ultrasonic treatment time and minimizing solvent degradation processes. It was demonstrated that the primary factor affecting cavitation efficiency was the solvent boiling temperature. Dynamic viscosity of the dispersion medium in the range of 0.8–2.5 Pa∙s, surface tension in the range of 36–43 N/m, and gas solubility (e.g., oxygen) in the range of 2.5–4.8 mmol/L had no significant impact on ultrasonic dispersion efficiency.

## Figures and Tables

**Figure 1 molecules-29-04846-f001:**
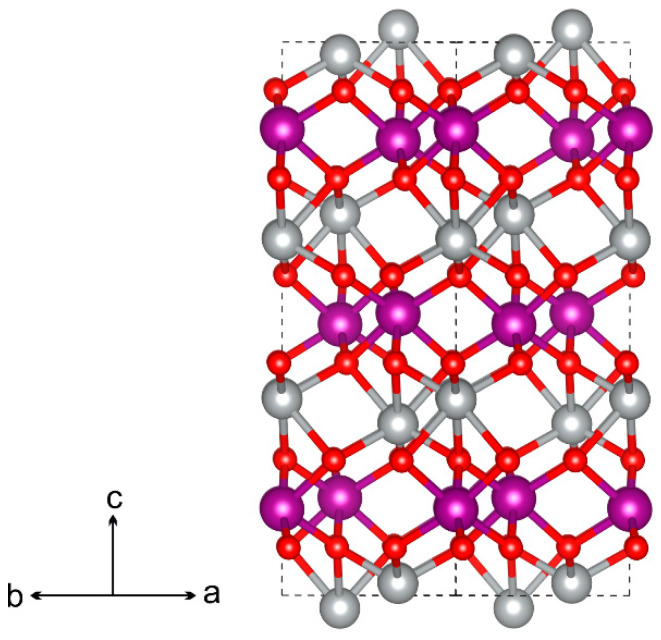
Atomic structure of NiMnO_3_. The unit cell is depicted by dashed lines. Nickel, oxygen, and manganese atoms are represented by gray, red, and violet colors, respectively.

**Figure 2 molecules-29-04846-f002:**
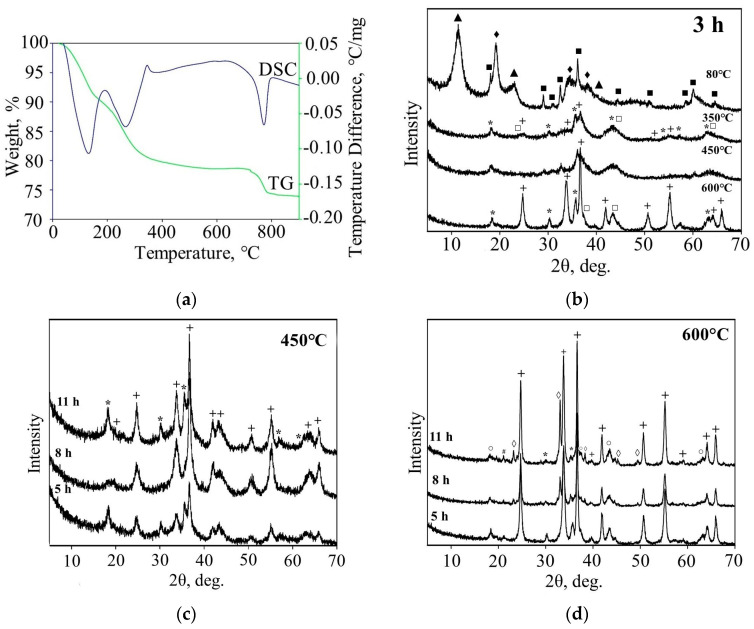
Thermogravimetry and differential scanning calorimetry (TG−DSC) curves (**a**) and X-ray diffraction pattern of the NiMnO_3_ precursor (**b**) annealed at different annealing temperatures (for 3 h) and annealed for different times: (**c**) at 450 °C; (**d**) at 600 °C +—NiMnO_3_, * NiMn_2_O_4_, □ NiO, ◊ Mn_2_O_3_, ○ Ni_6_MnO_8_, # MnO_2_, ♦ Ni(OH)_2_*NiOOH, ▲ MnOOH, ■ Mn_3_O_4_.

**Figure 3 molecules-29-04846-f003:**
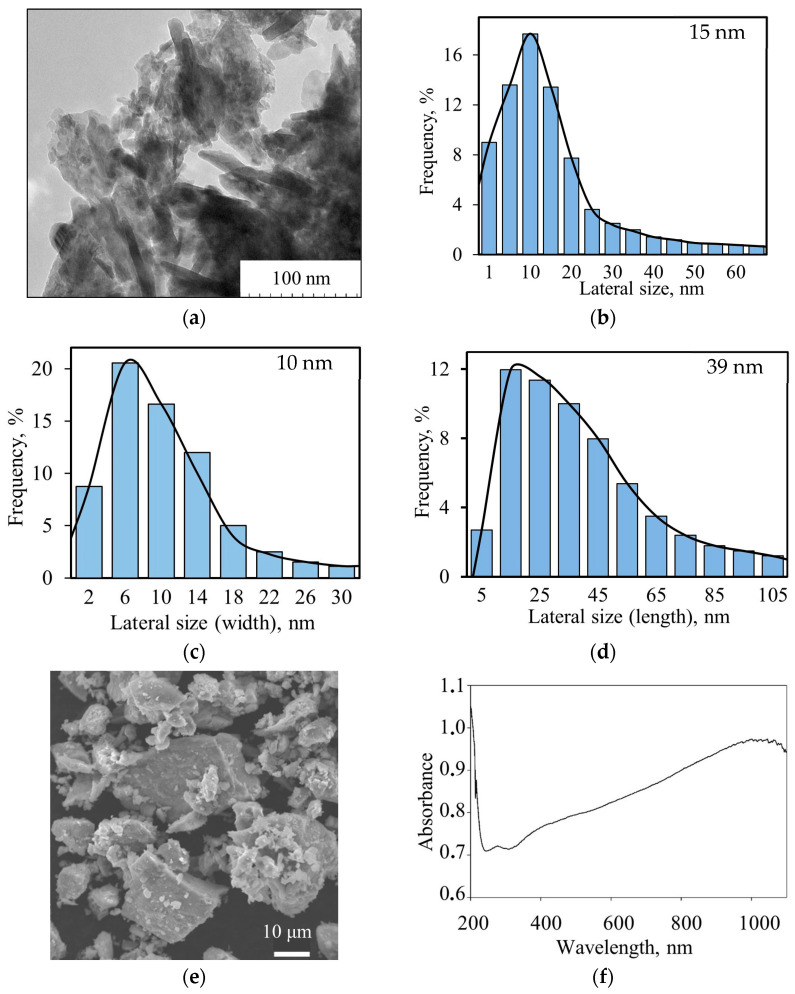
Morphological characterizations and physical properties of NiMnO_3_ nanoparticles: (**a**) low-resolution TEM image; (**b**–**d**) diagrams of the size distribution of spherical (**b**) and rod-shaped (**c**,**d**) nanoparticles; (**e**) SEM image; (**f**) UV–Vis spectra.

**Figure 4 molecules-29-04846-f004:**
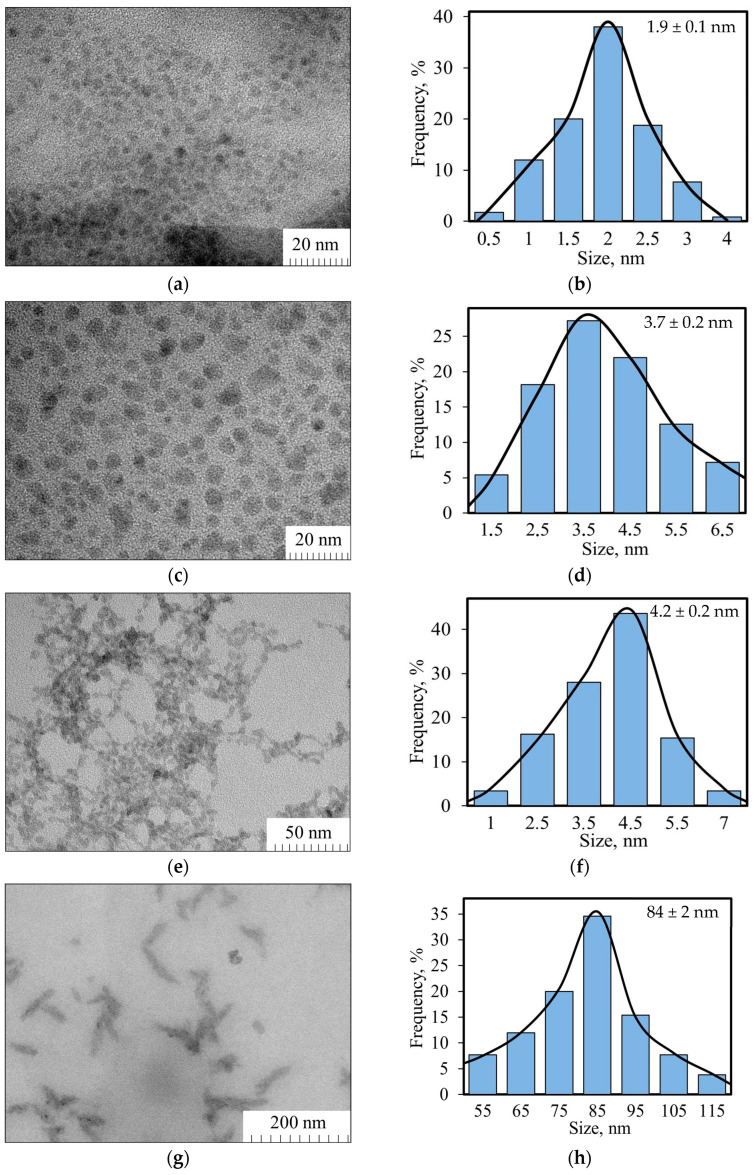
TEM images and diagrams of the size distribution of (**a**,**b**) NiMnO_3_-NMP; (**c**,**d**) NiMnO_3_-DMSO; (**e**,**f**) NiMnO_3_-DMF; (**g**,**h**) NiMnO_3_-H_2_O.

**Figure 5 molecules-29-04846-f005:**
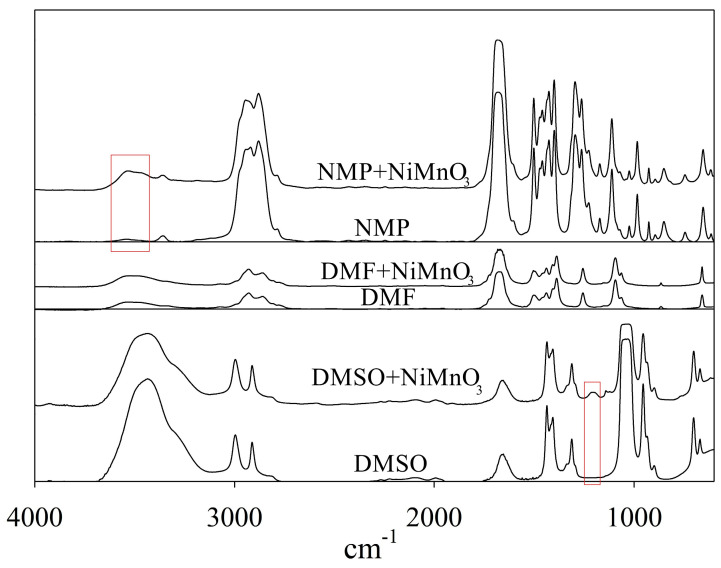
FT−IR-spectra of DMSO, DMF and NMP before and after US treatment of NiMnO_3_ in the spectral region of 4000–500 cm^−1^.

**Figure 6 molecules-29-04846-f006:**
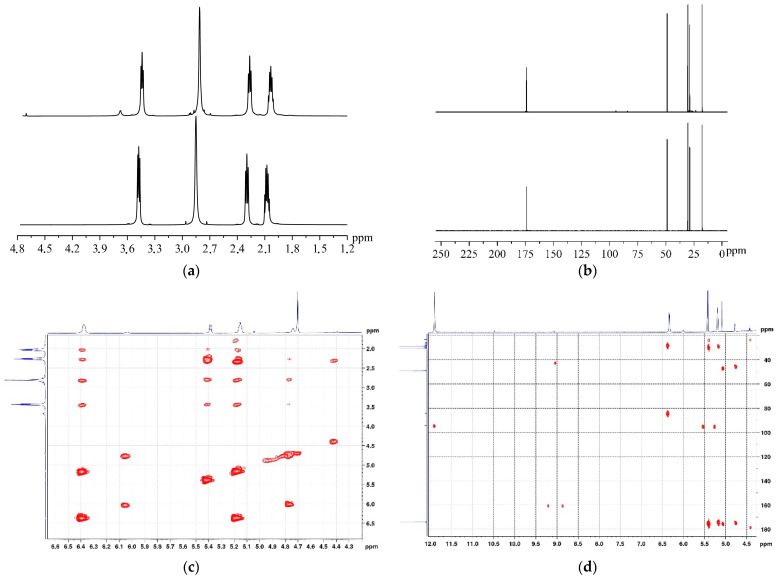
1H (**a**) and 13C (**b**) NMR spectra of NMP before (bottom line) and after (top line) ultrasound treatment; 1H-1H-DQF-COSY (**c**) and 1H-13C-HMBC (**d**) NMR spectra for NMP.

**Figure 7 molecules-29-04846-f007:**
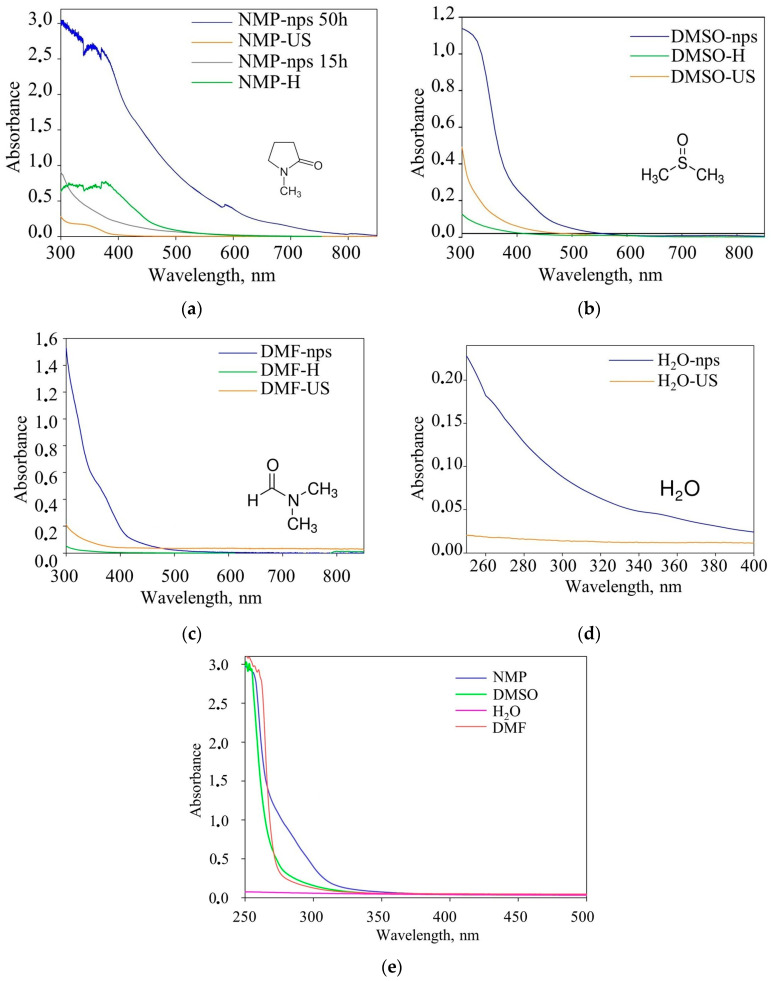
UV–Vis spectra of nanoparticles (nps) and solvents after heating (H) and ultrasound treatment (US) (recorded with pure DMSO or NMP as a reference) for (**a**) NMP, (**b**) DMSO, (**c**) DMF and (**d**) H_2_O; (**e**) UV–Vis spectra of pure solvents (recorded with empty cuvette as a reference).

**Figure 8 molecules-29-04846-f008:**
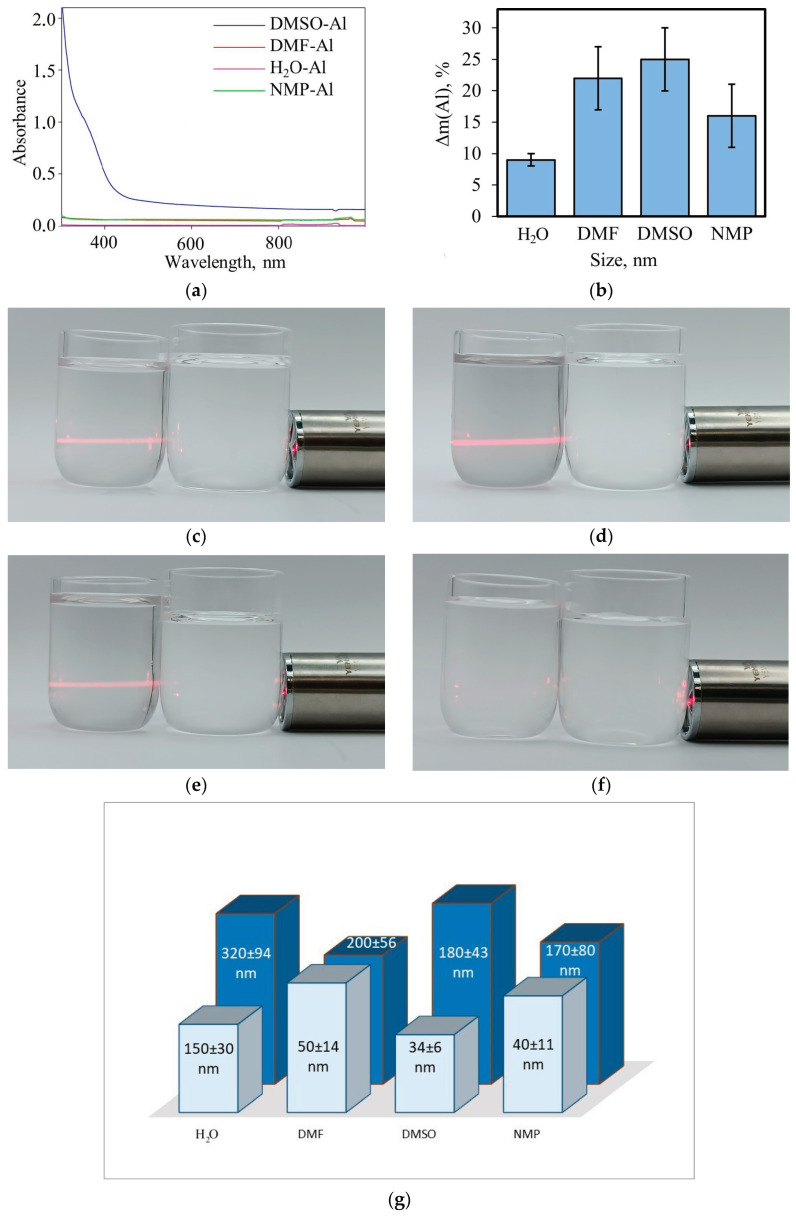
UV-vis spectra of Al sols (**a**). Histogram showing foil mass loss (**b**). Photos displaying Tyndall effect (**c**–**f**) for precipitates in: (**c**)—DMF, (**d**)—DMSO, (**e**)—NMP, (**f**)—H_2_O. Particle sizes in precipitates determined by DLS method (**g**).

**Table 1 molecules-29-04846-t001:** Influence of thermal treatment on the phase composition of the samples.

T, °C	t, h	NiMnO_3_	NiMn_2_O_4_	Mn_3_O_4_	NiO	Ni_6_MnO_8_	MnOOH	Ni(OH)_2_*NiOOH	Mn_2_O_3_	MnO_2_
80	3	-	-	33.0	-	-	57.0	10.0	-	-
350	3	-	-	69.5	30.5	-	-	-	-	-
450	3	23.2	61.5	-	15.3	-	-	-	-	-
	5	67.0	33.0	-	-	-	-	-	-	-
	8	100.0	-	-	-	-	-	-	-	-
	11	72.0	28.0	-	-	-	-	-	-	-
600	3	70.5	17.0	-	12.5	-	-	-	-	-
	5	89.2	5.3	-	-	5.5	-	-	-	-
	8	78.0	5.3	-	-	3.3	-	-	13.4	-
	11	67.7	2.6	-	-	4.9	-	-	22.3	2.5

**Table 2 molecules-29-04846-t002:** Cell parameters of the sample NiMnO_3_ obtained under optimal conditions.

Space Group Symmetry	Cell Parameters, Å		Cell Volume, Å^3^	Weighted Profile R-Factor, %	χ^2^
	a	c			
R-3	4.921 ± 0.002	13.658 ± 0.001	286.4 ± 0.3	12.3	1.566

**Table 3 molecules-29-04846-t003:** The influence of acoustic power absorbed by the solvent on the particle size of NiMnO_3_.

Parameters	NMP	H_2_O	DMSO	DMF
∆T, °C	5 ± 1	3 ± 1	5 ± 1	2 ± 1
P, W/cm^3^	1.1 ± 0.2	0.8 ± 0.3	0.7 ± 0.1	0.3 ± 0.1
d, nm	1.9 ± 0.1	84 ± 2	3.7 ± 0.2	4.2 ± 0.2

## Data Availability

The original contributions presented in the study are included in the article/Appendix A, further inquiries can be directed to the corresponding author.

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
