# Peer review of "Tailoring of Ultrasmall NiMnO3 Nanoparticles: Optimizing Synthesis Conditions and Solvent Effects"

_molecules, 2024, doi:10.3390/molecules29204846_

Round 1
Reviewer 1 Report
Comments and Suggestions for Authors
The presentation of this work is too unclear; I was unable, upon reading the manuscript, to determine the correlations, the objectives.
The Manuscript entitled « Tailoring of Ultrasmall NiMnO3 nanoparticles : Optimizing synthesis Conditions and Solvant Effects » is submitted as an Article to « Molecules » by S. Saikova and coll. As far as the title is concerned for this study, the manuscript is (to my point of view) not so understandable. Based on the results described, the obtained NPs (sephres and rods) are not so tailored, as the sizes are not well defined; the size distributions are quite large. For such data and for example concerning Figure 4, how do the authors deduce a size of “3,7 ±0,2 nm”? (and probably 3.7 ±0.2; please, check everywhere ..). The authors have to give more detailed comparisons with other procedure already described in the literature. To conclude, I am not convinced by the depicted procedure base of an ultrasonic destruction of material in order to finely tune the size of NPs.
Comments on the Quality of English LanguageCan be improved, but is not the main point (to my point of view), which makes the manuscript difficult to read.
Reviewer 2 Report
Comments and Suggestions for Authors
The manuscript titled “Tailoring of Ultrasmall NiMnO3 Nanoparticles: Optimizing Synthesis Conditions and Solvent Effects” investigates the synthesis of NiMnO3 via traditional coprecipitation methods and evaluates the effects of sonication under different solvents. While the study may be of interest to certain fields, several concerns and issues need to be addressed for reconsideration of its re-submission:
1. I have concerns about the conceptualization and methodology in this article, particularly the use of the coprecipitation method to fabricate NiMnO3. The authors refer to the as-prepared NiMnO3 as bulk materials, which contradicts their findings. Nanomaterials are defined as ranging from 1-100 nm, whereas bulk materials are larger. The authors' XRD results show a crystallite size of around 9.3 nm, and TEM measurements show nanorods with lengths of 40 or 39 nm, widths of 10 nm, and spherical particles with diameters of 15 nm—all within the 1-100 nm range. Therefore, these should be classified as nanoparticles, not bulk materials.
2. I don't see any synthesis described in section 2.2, unless the authors have a different definition of this concept. They mention using the coprecipitation method as a synthesis method, which is valid. However, they also discuss breaking down existing nanoparticles into smaller ones using sonication, which could alternatively be achieved using mechanical ball milling for bulk materials.
3. The description of the experiment design needs to be included, along with details on the data analysis performed, the statistical package used, and the total number of samples per treatment. The experimental design currently lacks clarity and completeness.
4. The Results section should be thoroughly discussed and compared with existing literature, as many studies have conducted similar research. These parameters have been comprehensively studied. Additionally, the phase change mechanisms need to be explained and supported with quantitative data in all sections.
5. The abstract suggests that the manuscript would cover magnetic and dielectric properties, sensor and supercapacitor applications, or catalytic water splitting. The absence of these topics weakens the manuscript and obscures the overall objective. The authors are advised to restructure and revise the manuscript to include these properties and applications for a more cohesive presentation.
6. The manuscript needs improvements in English language and writing quality. Additionally, there are discrepancies in the characterization techniques, with particle sizes being measured differently for the same technique across various sections and figures. Please revise these inconsistencies for clarity and accuracy.
Comments on the Quality of English LanguageThe manuscript needs improvements in English language and writing quality.
Reviewer 3 Report
Comments and Suggestions for Authors
In general, I recommend publishing the article in its current form. However, I can give some comments.
1. Line 306 (page 11) says Nickel nitrate (Ni(NO3)2·7H2O). It may be Nickel nitrate (Ni(NO3)2·6H2O) - check the water content. Nickel nitrate usually has six water molecules.
2. The method for synthesizing bulk NiMnO3 (lines 317-319) uses a stage of washing off impurities. How was the washing done and what is the degree of washing achieved?
3. X-ray diffraction data (Figure 2). Line 118: Calcination at 450 ℃ resulted in a spinel content of 64%, 23% NiMnO3 and 15% nickel 118 oxide. How was the quantitative calculation performed from X-ray diffraction patterns. 64+23+15=102 - needs to be adjusted so that the sum is 100%.
Comments on the Quality of English Language
No comments.
Reviewer 4 Report
Comments and Suggestions for Authors
The paper investigates the influence of thermal treatment parameters on the phase composition of products from alkali co-precipitation of nickel and manganese (II) ions, identifies optimal conditions for synthesizing NiMnO3, and systematically explores the solvent nature on the efficiency of ultrasonic dispersion of NiMnO3 nanoparticles. Therefore, I recommend its acceptance for publication in Molecules after the following minor revisions.
1. It is suggested to explore the influence of NMP concentration on NiMnO3 nanoparticle size.
2. Can the preparation of ultra-small size NiMnO3 by organic solvent on a large scale?
3. How to deal with the volatilization of organic solvents in the process of heating ultrasound?
4. The authors should discuss the effect of different organic solvents as ultrasonic dispersions on the structure of NiMnO3 phase.
5. The chemical transformation of NMP in the ultrasonic process promotes the dispersion of solid materials, and whether this transformation has functional group modification on the surface of solid materials.
6. The authors should explain in detail how the chemical changes of organic solvents during ultrasound affect the particle size.
7. The author should compare this work with the previous method of synthesizing ultra-small nanoparticles to illustrate its uniqueness.
Comments on the Quality of English Language
ok
Round 2
Reviewer 2 Report
Comments and Suggestions for Authors
The authors have addressed some of the concerns raised; however, there are still minor inconsistencies in the manuscript. Regarding the last comment, the authors were recommended to double-check the reported sizes in the discussion, figures, and conclusion to ensure consistency. For example, in Figure 3d, the length is listed as 40 nm in the discussion but 39 nm in the figure. Therefore, the authors should verify and correct these calculated parameters.
Additionally, the authors are recommended to calculate the crystallinity degree of the as-prepared and sonicated samples, as other relevant parameters can be obtained from XRD analysis.
Regarding comment 4, the authors should seriously discuss and compare their results with existing literature. They should include discussions on the effects of pH, calcination temperature, washing solvent, and ultrasound dispersion. These parameters are all studied in this research and need to be validated against previous studies.
Comments on the Quality of English LanguageMinor typos need to be corrected.